# SOMnibus: Recovering Underlying Sensitive Attributes with Self-Organizing Maps

**Joseph Bingham**
Department of Biology
The Technion – Israel Institute of Technology
`jbingham@campus.technion.ac.il`

**Netanel Arussy**
Department of Computer Science
The Technion – Israel Institute of Technology
`narussy@campus.technion.ac.il`

**Dvir Aran**
Department of Biology & Computer Science
The Technion – Israel Institute of Technology
`dviraran@technion.ac.il`

## Abstract

Unsupervised representation learning is often assumed to be benign with respect to sensitive attributes when those attributes are withheld from training. We challenge this assumption by demonstrating systematic *representation-level leakage* of ordinal sensitive attributes in purely unsupervised embeddings. Using **SOMnibus**, a topology-preserving method based on high-capacity Self-Organizing Maps, we show that attributes such as age and income emerge as dominant latent axes despite being explicitly excluded from the input. Across two large-scale real-world benchmarks, the World Values Survey and the Census-Income (KDD) dataset, SOMnibus recovers monotonic orderings aligned with withheld sensitive attributes, achieving Spearman correlations of up to $0.85$, while PCA and UMAP typically remain below $0.23$. Moreover, unsupervised segmentation of SOMnibus embeddings yields demographically skewed clusters, revealing downstream fairness risks in the absence of any supervised task. These results demonstrate that *fairness through unawareness* can fail at the representation level.

## 1 Introduction

Modern ML pipelines rely heavily on unsupervised representations (e.g., clustering, dimensionality reduction, embeddings) for exploration and downstream tasks Mehrabi et al. (2021). These representations are often assumed to be neutral when sensitive attributes (e.g., age, gender, income) are withheld.

This assumption—*fairness through unawareness* Dwork et al. (2012)—is known to fail in supervised settings, where models reconstruct sensitive attributes from proxies Kleinberg et al. (2016); Rabonato & Berton (2024). However, most work focuses on supervised predictors, leaving the extent to which *unsupervised* representations encode sensitive information underexplored Mehrabi et al. (2021).

This gap is critical: if embeddings organize data along sensitive axes, downstream uses inherit bias before any supervised task. Prior work shows demographic leakage in recommender embeddings Ekstrand & Kluver (2021), imbalance in clustering Chierichetti et al. (2017); Ghadiri et al. (2021), and sensitive structure in dimensionality reduction Peltonen et al. (2023).

In this work, we make four contributions:

**1)** We show that unsupervised representations encode sensitive attributes (age, income) with high fidelity, even when those attributes are excluded from training. **2)** We propose SOMnibus, based on high-capacity Self-Organizing Maps, as an unsupervised audit tool that reveals *global* sensitive structure, including monotonic orderings and gradients, that is attenuated or invisible to PCA or UMAP. **3)** We show that SOM-based representations leak substantially more sensitive information

Figure 1: The SOMnibus pipeline. (a) High-dimensional tabular data with sensitive attributes withheld. (b) A high-capacity Self-Organizing Map learns a topology-preserving discretization. (c) Activation patterns over the SOM lattice. (d) Observations embedded in 3D (BMU coordinates + activation energy), colored by withheld age group. For attributes with ordered structure, the SOM embedding recovers a monotonic axis aligned with the sensitive attribute.

than others, despite all methods being fully unsupervised. **4)** We argue that fairness interventions must operate at the representation level and include unsupervised components, motivating new auditing and mitigation strategies.

We study this phenomenon empirically on two large, real-world datasets: the World Values Survey (WVS) Haerpfer et al. (2020), a globally representative survey of political and moral values used in recent top-tier ML publications Adilazuarda et al. (2025); Li et al. (2024); Zhao et al. (2024), and the Census-Income (KDD) dataset cen (2000). In both cases, we withhold sensitive attributes entirely from the representation learning process and measure the degree to which they emerge as dominant latent structure.

Our results show that SOMnibus achieves Spearman correlations up to 0.85 between its learned embedding axes and withheld sensitive attributes, whereas PCA and UMAP typically remain below 0.23 (with a single exception reaching 0.31). Moreover, unsupervised segmentation of SOMnibus embeddings produces clusters with substantial demographic skew, demonstrating that fairness risks arise even in the absence of any supervised objective.

## 2 METHOD: SOMNIBUS AS A TOOL

We position SOMnibus not as a solution to fairness, but as a *lens* that makes sensitive structure in unsupervised representations explicit. The method reveals whether and how strongly a representation encodes withheld sensitive attributes.

**Self-Organizing Map Representation** In contrast to conventional SOM usage for two-dimensional visualization, we employ a map with a large number of neurons ($K = 5 \cdot N^{0.54}$). This scaling follows the heuristic proposed by Vesanto and Alhoniemi Vesanto & Alhoniemi (2000), which balances representational capacity against computational cost. The resulting lattice allows the SOM to approximate the data distribution at finer granularity than is typical in visualization applications.

**Auditing via Correlation Analysis** We assess alignment between the learned representation and the withheld sensitive attribute $s$ by computing Pearson and Spearman correlation coefficients between $s$ and each of the three embedding dimensions independently. We report the maximum absolute correlation across dimensions. This maximum captures the worst-case leakage: if *any* axis of the embedding is aligned with the sensitive attribute, the representation poses a fairness risk regardless of the other axes. Both Pearson (linear) and Spearman (monotonic) coefficients are reported to capture different forms of dependence. This evaluation is *purely diagnostic*: $s$ is never used for training, embedding, or extraction.

## 3 RESULTS

### 3.1 DATASETS

**World Values Survey (WVS).** The WVS Haerpfer et al. (2020) is a globally representative dataset of political and moral values. We use five countries—Canada ($N = 4,018$), Romania ($N = 3,200$), Germany ($N = 1,528$), China ($N = 3,036$), and USA ($N = 2,609$)—with **age** as the withheld attribute. We restrict to 22 ethics- and values-related questions with low correlation ($|\rho| \leq 0.09$).

**Census-Income (KDD).** The Census-Income dataset cen (2000) contains 299,285 records with 40 demographic and economic features. We treat **age**, **income**, and **capital gains** as withheld sensitive attributes, and use all features convertible to numeric form, without filtering for correlation.

## 3.2 EXPERIMENTS

**Sensitive Attribute Recoverability.** For each method (PCA, UMAP, SOMnibus), we compute embeddings with sensitive attributes withheld and report the maximum Pearson and Spearman correlation between any embedding axis and the attribute, measuring leakage.

**Global Ordering Emergence.** We extract a dominant 1D axis from each representation (via clustering for SOMnibus) and compute its Spearman correlation with the sensitive attribute, testing for monotonic global ordering.

## 3.3 SENSITIVE ATTRIBUTE LEAKAGE

Table 1 presents the core results. Across all datasets and sensitive attributes, SOMnibus achieves substantially higher correlations with the withheld sensitive attribute than either PCA or UMAP. On the WVS dataset, SOMnibus achieves Spearman correlations of 0.43–0.85 with age across five countries, while PCA and UMAP achieve at most 0.31. On the Census dataset, SOMnibus achieves Spearman correlations of 0.83 (age), 0.69 (income), and 0.43 (capital gains), compared to maxima of 0.21, 0.21, and 0.11 for the baselines.

A natural question is whether SOMnibus's higher correlations reflect greater overall capacity rather than preferential leakage. While its larger capacity ($K^2$ prototypes) likely preserves more structure, the gap is substantial (3–8× over baselines). Distinguishing amplification from proportional preservation requires evaluating non-sensitive reconstruction, which we leave to future work.

## 3.4 GLOBAL ORDERING EMERGENCE

Figure 2 shows the SOM embedding for WVS, colored by age (withheld during training). Age groups form a clear monotonic ordering, while non-respondents (cyan) cluster near the center, consistent with no age signal. This demonstrates strong representation-level encoding of age.

To quantify this, we extract the dominant 1D axis and compute its Spearman correlation with age. On WVS Canada, SOMnibus achieves $\rho = 0.82$, with similar results across datasets (0.60–0.82). In contrast, PCA and UMAP show substantial overlap between age groups and no recoverable structure.

Table 1: Correlation (Pearson/Spearman) with sensitive attribute.

| Data | Dom. | PCA | UMAP | SOM |
|------|------|-----|------|-----|
| WVS | CA | -0.25/0.22 | -0.31/0.31 | 0.75/0.85 |
| | RO | 0.06/0.04 | -0.06/-0.05 | 0.66/0.59 |
| | DE | -0.16/-0.18 | 0.17/0.19 | 0.79/0.73 |
| | CN | -0.22/-0.22 | 0.21/0.20 | 0.80/0.58 |
| | US | -0.20/-0.23 | 0.02/0.02 | 0.80/0.52 |
| Census | Age | 0.17/0.11 | 0.08/0.09 | 0.50/0.83 |
| | Inc | -0.31/0.21 | 0.07/0.07 | 0.48/0.69 |
| | Gain | -0.23/-0.11 | 0.01/0.002 | 0.34/0.43 |

SOMs for WVS are trained per country, while a single model is used for Census. Despite low correlations between sensitive attributes (e.g., age–income $-0.11$), SOMnibus recovers multiple attributes, though not uniformly (e.g., weaker for capital gains). Overall, these results show that SOMs can recover sensitive attributes from non-sensitive data across domains, highlighting persistent representation-level bias.

## 4 CONCLUSION

Unsupervised learning is not neutral: representations can encode sensitive attributes as geometric structure, with topology-preserving methods like SOMs particularly effective at revealing this. Across two real-world datasets, SOMnibus recovers monotonic orderings aligned with age and income that are not captured by PCA or UMAP, and yields demographically skewed clusters. These results imply that fairness auditing must extend to unsupervised representations. We advocate routine leakage audits (e.g., correlation and ordering analysis) prior to deployment, with SOMnibus providing a practical tool to expose such risks.

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

## A    SOMs Overview

We map the input data into a two-dimensional discrete latent space using a Self-Organizing Map (SOM). Let $X \in \mathbb{R}^{N \times d}$ denote the input matrix. The SOM consists of a square lattice of $K \times K$ units, where each unit $j$ is associated with a prototype vector $\mathbf{w}_j \in \mathbb{R}^d$.

For each input observation $\mathbf{x}_i$, the best-matching unit (BMU) is defined as

$$j^* = \arg\min_j \|\mathbf{x}_i - \mathbf{w}_j\|.$$

Prototype vectors are updated according to

$$\mathbf{w}_j(t+1) = \mathbf{w}_j(t) + \alpha(t) \, h(j, j^*, t) \, (\mathbf{x}_i - \mathbf{w}_j(t)),$$

where $\alpha(t)$ is a learning rate and $h(\cdot)$ is a Gaussian neighborhood function defined over the lattice topology. This competitive-learning rule induces a topology-preserving discretization of the data manifold: nearby neurons in the lattice specialize to similar regions of the input space.

## B    Implementation Details

All experiments are implemented in Python using standard scientific computing libraries. Hyperparameters for the SOM are fixed across all experiments and are not tuned using sensitive attribute labels: $\sigma = 0.7$, learning rate $= 0.75$, with lattice dimension $K = 5 \cdot N^{0.54}$, where $N$ is the number of inputs. The selection criteria implemented for determining questions to be used in experiments was such that the dataframe does not contain missing values. This design ensures that reported correlations reflect emergent structure rather than supervised optimization. Since both SOM and UMAP involve stochastic initialization, all reported results are averaged over five independent runs; standard deviations were below 0.03 for all Spearman correlations, indicating stable results.

## C  SUPPLIMENTAL RESULTS

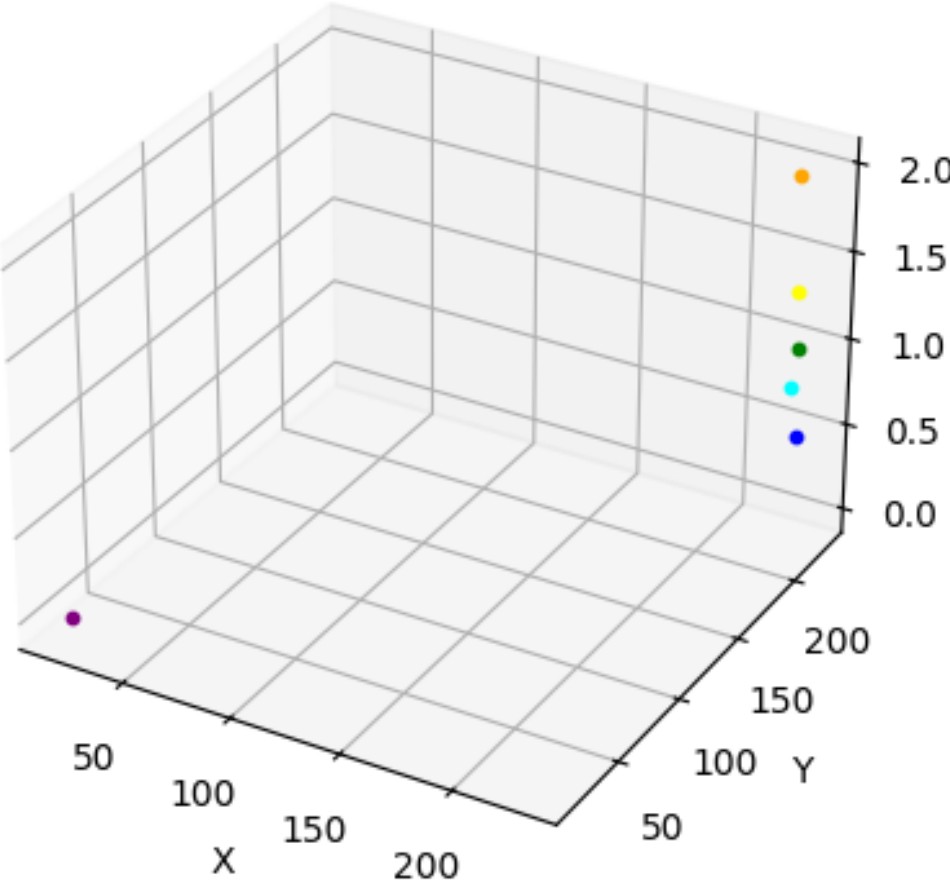

Figure 2: Sensitive attribute emergence in the SOM embedding from centroids formed from SOM clustering. Age groups (withheld from training) form a clear monotonic ordering along the learned latent surface. Colors: purple (youngest) → blue → green → yellow → orange (oldest). Cyan: non-respondents, distributed near centered.

