# OpenReview forum: "SOMnibus: Recovering Underlying Sensitive Attributes with Self-Organizing Maps"
_ICLR.cc/2026/Workshop/AFAA — AFAA 2026 Poster_

### Official Review · Reviewer_882Z · 2026-02-19
**On Dimensionality Reduction and Biases**

**Rating:** 2
**Confidence:** 5

**Summary:**

The objective of this paper is to study biases in unsupervised representations before moving on to downstream tasks. They first introduce a dimensionality-reducing transformation that they call SOMnibus, which is a topology-preserving method based on high-capacity Self-Organizing Maps. Next, they show that using such a transformation, one can have significant correlation (both Pearson and Spearman) with sensitive attributes as compared to other dimensionality-reduction techniques such as PCA.

**Strengths:**

The transformation technique seems interesting.

It is good to note that different transformations can differ on how much bias they preserve. This establishes the point that auditing unsupervised representations using dimensionality reduction would be sensitive to the choice of transformation.

The writing is easy to follow.

One of the datasets considered, i.e.,  World Values Survey (WVS) is a multi-class dataset.

**Weaknesses:**

The first contribution states that biases do exist in the hidden/unsupervised representations. This contribution in itself is not so surprising, and need not be stated as a contribution. If there were no biases, indeed downstream tasks would already be bias free.

Only two datasets are considered, and only two baselines (alternate dimensionality reduction techniques) PCA and UMAP are considered.

While correlation is an interesting metric for the unsupervised representations, its impact would mostly be relevant for downstream tasks audited using statistical parity. Other than correlation, could there be alternate metrics that would be relevant for equalized-odds-style auditing? I.e., is it necessary to avoid all correlation with protected attributes in the unsupervised representation stage, or could some correlations be preserved?

While the initial idea is promising, I think the paper needs more extensive validation for the Main Papers Track. Could be considered for a Tiny/Short paper.

---

### Official Review · Reviewer_NWYC · 2026-02-20
**Review: SOMNIBUS: RECOVERING UNDERLYING SENSITIVE ATTRIBUTES WITH SELF-ORGANIZING MAPS**

**Rating:** 3
**Confidence:** 2

**Summary:**

This paper investigates whether unsupervised representations encode withheld sensitive attributes, challenging the "fairness through unawareness" assumption. The authors propose SOMnibus, a method based on high-capacity Self-Organizing Maps that maps inputs to a 3D embedding and then measures correlation between embedding axes and withheld  attributes . Evaluated on the World Values Survey and Census-Income datasets, SOMnibus achieves Spearman correlations up to 0.85 with age. Overall, the paper addresses a timely and practically important question — whether unsupervised embeddings inherit fairness risks.

**Strengths:**

The magnitude of the gap between SOMnibus and the baselines is the paper's strongest asset, and the framing of the tool as an *auditing* mechanism rather than a mitigation is appropriately scoped.

1. **Important and well-scoped research question.** The paper draws attention to a genuine blind spot in the fairness literature:  unsupervised representations used upstream can already encode sensitive structure.
2. **Empirical gap over baselines.** The difference in Spearman correlation  is substantial and consistent across datasets. The SOM-based approach reveals structure that PCA and UMAP genuinely miss, which makes a compelling case
3. **Thoughtful dataset selection.** Using the WVS alongside Census-Income provides diversity in both domain and data characteristics. The WVS subset is carefully chosen to include only questions with absolute mean correlation to age ≤ 0.09
4. **Clear presentation of the method.** The pipeline  is simple and reproducible. The scaling heuristic  fixed hyperparameters, and averaged results over 5 runs  all support reproducibility.

**Weaknesses:**

1. **Capacity confound is not resolved.** The authors acknowledge that SOMnibus uses $K^2$ prototype vectors while PCA and UMAP produce 2–3D embeddings, making the comparison asymmetric. The paper concedes this but defers measuring non-sensitive reconstruction quality to future work.
2. **Narrower baseline set.** PCA and UMAP are reasonable starting points, but the paper does not compare against probing classifiers (the standard auditing method acknowledged in Section 2.2) or other nonlinear methods with comparable capacity.
3. **Restriction to ordinal attributes.** The method relies on Pearson/Spearman correlation, which limits applicability to ordinal sensitive attributes. The authors note this in the limitations.

---

### Official Review · Reviewer_1Yhu · 2026-02-20
**Self-Organizing Maps reveal leakage of withheld sensitive attributes in unsupervised setting**

**Rating:** 3
**Confidence:** 3

**Summary:**

The paper presents a pipeline named SOMnibus that reveals that withholding sensitive attributes does not necessarily prevent them from leaking in learned embeddings.
The authors obtain an embedding using SOM on non-sensitive features and quantify leakage by evaluating the degree to which the withheld sensitive information correlates with these embeddings.
Significant correlations are obtained in real-world experiments.
The authors conclude that unsupervised representations can lead to alignment with withheld sensitive information.

**Strengths:**

S1. The paper addresses a practical point that removing sensitive columns does not necessarily remove sensitive information from learned representations.

S2. The method is interpretable, as SOM provides a structured grid and a notion of where the embedded points are located.

**Weaknesses:**

W1. My main concern is over the leakage metric; it takes the maximum correlation over the embedded axes. This can inflate the reported correlations under chance.

W2. The scope is a bit limited as the method only deals with tabular data with ordinal attributes. It is unclear how this will scale with sensitive features, which are primarily categorical.

W3. To some extent, the comparisons are not completely fair: UMAP is run in 2D, and PCA is treated differently, while SOM uses a very large lattice. The higher leakage might indicate that SOMnibus retains more information overall.

---

### Meta-Review · Area_Chair_YE5c · 2026-02-23

**Recommendation:** Tiny/Short Papers Track
**Confidence:** 3

**Metareview:**

The paper proposes a pipeline called SOMnibus that uncovers the influence of underlying attributes even when they are removed from the data. The motivation is good and the applications are impactful. All reviewers found the ideas interesting and practical, but had foundational concerns regarding evaluation: the data sets used (while very well-chosen) seem limiting and there are two baselines that could use extensions, as one of the reviewers pointed. Given limited but good contributions, I recommend the paper be presented as a poster in the tiny papers track.

---

### Decision · Program_Chairs · 2026-03-02

**Decision:**

Accept (Poster)

**Comment:**

The paper was originally submitted under the Main Track. While the paper is not ready to be accepted as a Main Track Paper, we find the work promising, and are giving an opportunity to the authors to instead get accepted in the Tiny/Short Track. Please submit a camera-ready version of up to 3 pages to comply with the Tiny/Short Paper requirements (more instructions on how to submit the camera-ready version will follow soon). The authors can also decide to withdraw if they prefer to not be accepted under the Tiny/Short Track.